

# Statistical Appraisal of Geothermal Heat Flow Observations in the Arctic

Judith Freienstein[1], Wolfgang Szwillus[1], Agnes Wansing[1], and Jörg Ebbing[1]

[1]Institute of Geosciences, Kiel University, Otto-Hahn-Platz 1, 24118 Kiel

**Correspondence:** Judith Freienstein (judith.freienstein@ifg.uni-kiel.de)

**Abstract.** Geothermal heat flow is an important boundary condition for ice sheet, affecting for example basal melt rates, but for ice covered regions we only have sparse heat flow observations with partly high uncertainty. In this study, we first investigate the agreement between such point-wise heat flow observations and Solid Earth models, applying a 1D steady state approach to perform a statistical analysis for the entire Arctic region. We find that most of the continental heat flow observations have

5 a high reliability and agreement to Solid Earth models, except a few data points, as for example the NGRIP point in Central Greenland.

For further testing, we perform a conditional simulation with focus on Greenland, in which the local characteristics of heat flow structures can be considered. Simple kriging shows that including or excluding the less reliable NGRIP point has a large influence on the surrounding heat flow. The geostatistical analysis with the conditional simulation supports the assumption that

NGRIP might not only be problematic for representing a regional feature but likely is an outlier. Basal melt estimates show that such a local spot of high heat flow results in local high basal melt rates, but leads to less variation than existing geophysical models.

## 1 Introduction

Geothermal heat flow (GHF) is a key factor of Solid Earth-cryosphere interaction. Under ice-covered regions, such as Greenland, GHF is a boundary condition for ice sheet dynamics (Karlsson et al., 2021; Goodge, 2018). Karlsson et al. (2021) state that GHF can contribute up to 25 % of total basal melt rates. As shown in McCormack et al. (2022), locally high heat flow can have a larger impact on ice sheet dynamics compared to a regionally higher value. GHF itself is influenced by the Solid

Earth, reflecting first of all the thickness of the lithosphere (Lösing and Ebbing, 2021), but hydrological processes (Gooch et al., 2016) or crustal heat production variations (Bons et al., 2021) also contribute, making heat flow for continental settings highly variable (Reading et al., 2022). Maps of GHF are often based on interpolation of sparse observations, so isolated points might distort the distribution. Geothermal heat flow is complicated to measure directly. Borehole measurements are expensive



and therefore sparse in large parts of the Arctic (and Antarctic), which is covered with ice and snow most of the year. Addi-
tionally, observations are concentrated in areas of economic interest or that are easily accessible (Stål et al., 2022). Therefore,
Arctic heat flow observations are distributed very heterogeneously with dense data coverage in regions around the mid-oceanic
ridge, Scandinavia and the north of Canada and poor data coverage in Siberia, Greenland and the Arctic Ocean north of Alaska
(Lucazeau (2019), Figure 2).

Ice temperature profiles present another option to estimate GHF in glacial areas (e.g. Dahl-Jensen et al. (2003)). If the borehole
reaches the ice-bedrock interface, GHF can be estimated using models of heat transport in the column of ice (e.g. Weertman
(1968); Rasmussen et al. (2013)). However, some boreholes have not reached bedrock, so the ice temperature profiles need to
be extrapolated (e.g. Kinnard et al. (2006); Buchardt and Dahl-Jensen (2007)), leading to large uncertainties for estimated heat
flow values. The NGRIP point in central Greenland is a particularly notorious example (Buchardt and Dahl-Jensen, 2007), with
a wide range of values between $63\,\mathrm{mWm^{-2}}$ (for example Martos et al. (2018)) and $970\,\mathrm{mWm^{-2}}$ (Smith-Johnsen et al., 2020)
being suggested in the literature. The latter estimate is highly unlikely but even the most conservative estimates well exceed
the mean of Greenland with $60\,\mathrm{mWm^{-2}}$ (Colgan et al., 2022).

Estimating the heat flow from geophysical data gives the possibility to overcome the sparseness. Curie depth estimates based
on magnetic data are a classical tool to infer heat flow. For Greenland, heat flow was derived from Curie depth estimates based
on satellite magnetic (Fox Maule et al., 2009) or aeromagnetic (Martos et al., 2018) compilations. Thermal models of the entire
lithosphere can also be constrained by a variety of geophysical data sets (i.e. gravity, surface wave data), but these models
typically lack lateral resolution within the crust (Afonso et al. (2019), Pasyanos et al. (2014), Fullea et al. (2021)).

A more geostatistical approach is to compare proxies in a region with poorly known heat flow with similar proxies in regions
with good coverage. Upper mantle seismic velocity was one of the first proxies used to infer GHF (Shapiro and Ritzwoller,
2004) , while Artemieva (2019) applied a thermal isostasy model. More recently, machine learning algorithms (specifically
random forest regressors) have been used to predict geothermal heat flow based on a variety of geographical/geophysical prox-
ies (Colgan et al., 2022; Rezvanbehbahani et al., 2017; Lösing and Ebbing, 2021). See Colgan et al. (2022) for an extended
discussion on GHF models for Greenland. However, such heat flow maps can only present the regional heat flow as they are
limited by the availability and resolution of data used for analysis. The non-linear optimization heuristic used in the machine
learning techniques is also highly sensitive to isolated data points. Colgan et al. (2022) and Rezvanbehbahani et al. (2017)
study this by omitting or varying the estimated GHF value for individual data points, respectively and find that particularly the
NGRIP point presents a challenge, being a highly uncertain and isolated measurement. Without additional information, local
structures and regional features cannot be distinguished based on sparse point measurements. Heat flow anomalies can be as
small as a few tens of kilometers due to shallow crustal heat production and the effect of subglacial topography (Reading et al.,
2022). Thus, interpolation (or random forest regression) of GHF observations is prone to large biases if local anomalies are
mistaken for regional features. Nevertheless, local GHF anomalies are crucial for cryosphere-solid earth interaction. (McCor-
mack et al., 2022)

In this study, we approach the question of local vs. regional effects on GHF from two angles. Firstly, we evaluate a database
of GHF measurements by testing each individual measurement's consistency with a lithospheric temperature model based on



estimates of Moho and LAB (Lithosphere-Asthenosphere Boundary) depth. Secondly, we use geostatistical analysis and con-
ditional simulation to investigate the spatial scale of heat flow in Greenland. Our results can help to decide whether to exclude
points for interpolation and machine learning on a regional scale or in regions with sparse data, as they are not trustworthy for
such applications.

## 2 Methods

For any heat flow observation, it is unknown whether the observed value reflects the regional setting or a local anomaly.
Assuming that regional structures are in agreement with LAB and Moho depth models, it should be possible to find a set
of thermal parameters (heat conductivity and heat productivity), such that GHF can be predicted from stationary 1-D heat
flow modeling (Lösing et al., 2020; Furlong and Chapman, 1987; Artemieva and Mooney, 2001). If no combination of the
parameters within their given ranges lead to an agreement, the GHF observation should be considered erroneous or a local
anomaly. For example, points in areas of exceptionally local high heat production from radiogenic sources (Bons et al., 2021),
should lead to an incompatibility between the lithospheric model and GHF data.

We assume that geophysical LAB depth can be seen as a representation of the large scale lithospheric temperature field, so
we compare the predicted temperature at the LAB to an assumed LAB temperature of $1315\,°C$. If the temperature deviation
surpasses a threshold of $100\,K$, we assume that the corresponding GHF observation probably is locally influenced and therefore
cannot resolve the regional assumptions of the geophysical models. Choosing such high deviation we take uncertainties from
the used models for Moho and LAB into account.

We assume vertical heat flux within the lithosphere, which is a common assumption at least for the continental domain (Afonso
et al., 2013; Lösing et al., 2020). Furthermore, the lithospheric columns are assumed to be in thermal equilibrium, resulting in
the temperature equation:

$$k_1 \frac{\partial^2 T}{\partial z^2} = h(z), \tag{1}$$

with the crustal thermal conductivity $k_1$, the temperature $T$, the depth $z$ and the heat productivity $h$.

Assuming no heat generation in the lithospheric mantle, the temperature increases linearly with depth, so that the temperature
in the lithospheric mantle at a given depth can be calculated with

$$T(z) = T(M) + \frac{q_D}{k_2}(z - M), \tag{2}$$

with the Moho depth $M$, the mantle heat flux $q_D$ and the mantle thermal conductivity $k_2$.

Heat production $A$ is assumed constant with depth, following Lösing et al. (2020). The heat flux $q$ at a certain depth is then

$$q(z) = q_0 - Az, \tag{3}$$

where $q_0$ is the heat flux at the surface. When calculated at $z = M$ we get $q_D$.

For the temperature distribution in the crust we get

$$T(z) = T(0) + \frac{zq_0 - \frac{1}{2}Az^2}{k_1}. \tag{4}$$



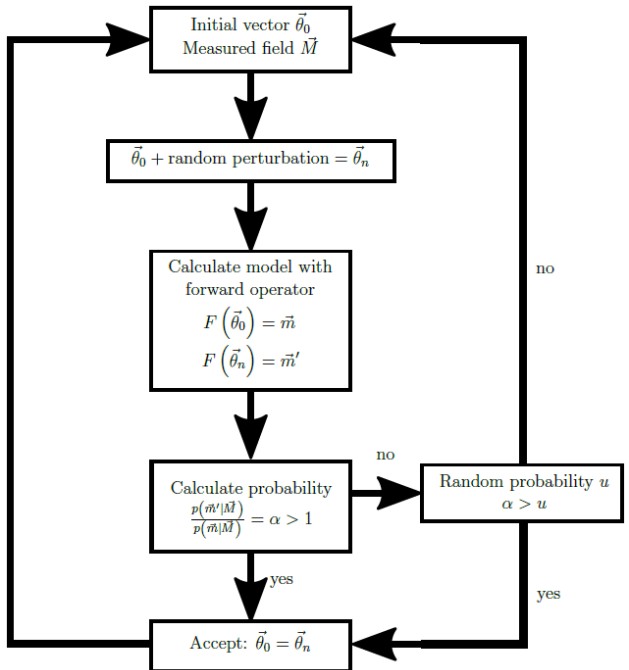

**Figure 1.** Scheme of the MCMC algorithm as described in the text.

Further derivations can be taken from Lösing et al. (2020).

The Moho temperature can be determined by means of this temperature distribution, thus a temperature at a certain depth in the lithospheric mantle is:

$$T(z) = T(0) + \frac{Mq_0 - \frac{1}{2}AM^2}{k_1} + \frac{q_D}{k_2}(z - M).$$ 

(5)

We now can fit the heat flow observations to the temperature profile based on geophysical data and Eq. (5) using a Bayesian inversion coupled with a Monte Carlo Markov Chain algorithm. This approach is based on the method presented in Lösing et al. (2020). The goal is to adjust a parameter vector so that the calculated model corresponds as closely as possible to the given model $M$, here the LAB temperature. For this purpose, Eq. (5) is to define a forward operator $F(\Theta)$ that calculates the temperature at the LAB for a given parameter vector $\Theta$. As a result we also get estimates for the crustal and mantle thermal
conductivity $k_1$ and $k_2$ and crustal heat production $A$. The principle of the algorithm is explained in Fig. 1.

First, initial values for the thermal parameters are stored in a vector $\Theta$. The forward operator now takes this vector and calculates a corresponding proposed model $F(\Theta) = m$. A standard deviation of $\sigma_T^2 = 100\,K$ is allowed, due to uncertainties in the Moho and LAB depth models, given by the relationship $\sigma_z = \sigma_T k/q$ resulting in a few tens of kilometers uncertainty for the input depths. For each iteration, we change the initial parameter vector by adding random perturbations. Based on the





fit of this standard deviation and prior information, a likelihood is assigned to each of the resulting parameter vectors:

$$L(\boldsymbol{M}|\boldsymbol{\Theta}) = \prod_{i=1}^{N} \frac{1}{\sqrt{2\pi\sigma_M^2}} \exp{-\frac{(F(\boldsymbol{\Theta})_i - \boldsymbol{M}_i)^2}{2\sigma_M^2}}. \tag{6}$$

If the proposed model has a higher probability than the current model, its parameter vector will be used as the new initial parameter vector. If the probability is lower, the proposed model can still be accepted with a probability $P_{\text{new}}/P_{\text{old}} > u$ where $u$ is a uniformly distributed random number between 0 and 1. This prevents being caught in local minima (Lösing et al., 2020). To deliver representative results, a certain number of iterations and burn-in iterations are needed. In our case we found 10,000 iterations with 5,000 discarded burn-in iterations sufficient. To eliminate random fluctuations, the mean of the iterations gives the resulting parameter vector. As prior information, the allowed range of parameters is given. The ranges base on typical physical properties relying on studies for regional thermal parameter ranges (e.g. Artemieva and Mooney (2001); Furlong and Chapman (1987); Jaupart and Mareschal (2014)). Although higher crustal heat production up to $5\,\mu\mathrm{Wm}^{-3}$ can be assumed for some structures, we use the average values for regional structures. The corresponding ranges for the parameters, the start values and allowed maximum step sizes within an iteration are shown in Table 1. We assume the surface temperature as $0\,^{\circ}\mathrm{C}$ and the LAB temperature as $1315\,^{\circ}\mathrm{C}$ (according to Lösing et al. (2020)).

**Table 1.** Prior information for the inversion: initial value, range, and step size for each iteration.

| Parameter | initial value | initial range | step size |
|---|---|---|---|
| $k_1$ in $\mathrm{Wm}^{-1}\mathrm{K}^{-1}$ | 2.2 | [1.0, 3.0] | 0.5 |
| $k_2$ in $\mathrm{Wm}^{-1}\mathrm{K}^{-1}$ | 3.0 | [2.5, 4.0] | 0.5 |
| $A$ in $\mu\mathrm{Wm}^{-3}$ | 0.7 | [0.25, 1.75] | 0.375 |

## 2.1 Kriging Interpolation and Conditional Simulation

Kriging interpolation of irregularly spaced data sets provides estimates with confidence intervals (Cosentino et al., 2023). Here, we assume the mean value $m_0$ as given and constant, so "simple Kriging" results (Chiles and Delfiner, 1999):

$$Z^* = m_0 + \sum_{\alpha} \lambda_\alpha (Z_\alpha - m_\alpha), \tag{7}$$

with the Kriging estimator $Z^*$, the weights $\lambda_\alpha$, the given observations $Z_\alpha$ and the mean of the observations $m_\alpha$. The weights are adjusted so that the resulting estimator (Eq. (7)) is unbiased and the error variance minimal (Cosentino et al., 2023). A crucial parameter for kriging interpolation is the covariance function, which determines how quickly the weighting decreases with distance (Chiles and Delfiner, 1999). Here, we use the Gaussian covariance model (Webster and Oliver, 2007):

$$\gamma(r) = \sigma^2 \left( 1 - \exp\left[ -\left( s \cdot \frac{r}{\ell} \right)^2 \right] \right) + n, \tag{8}$$



in which $r$ represents the distance of the points, $\sigma^2$ the variance of the model, $s = \sqrt{\pi}/2$ the rescaling factor, $\ell$ the length scale and $n$ the nugget.

The correlation length scale $\ell$ can be estimated using the semivariogram representing the dissimilarity of pairs of points at a certain distance. Closer points tend to be more similar, increasing the distance of the points, the dissimilarity usually also increases. The correlation length is defined at which the dissimilarity reaches a certain threshold. The nugget describes small-scale effects, when points with very small distances have an offset to the original point (Wackernagel, 1998). A minimum number of 100 points should be used for a representative variogram where we can define distances that resolve the resolution

of the data as well as getting an accurate estimate for the mean semi variance (Cosentino et al., 2023).

Kriging interpolation is based on a geostatistical approach, such that the interpolation result is a multivariate normal distribution. The mean (i.e. expected) value at each point is typically taken as the result and the pointwise standard deviation as uncertainty. However, this is much smoother than any realization of the actual solution, because it neglects the correlation of errors at different locations. Sparse and uneven distributed data by itself can lead to overestimation of the correlation lengths,

which results in unrealistic uncertainty estimates (Chiles and Delfiner, 1999; Hadavand and Deutsch, 2020). Assuming that there is a geological region similar to the study area but with higher data coverage, we can use the covariance function (especially the correlation length) estimated for that region in the study area instead. Conditional simulation can be used to generate realizations that show possible smaller scale variations. We use this to assess the likelihood of the NGRIP result being a local anomaly or measurement error. The conditional simulation is based on a two-stage Kriging evaluation combined with an

unconditional simulation. First, the given points are interpolated using the Kriging method ($Z^*(x)$). Then a sample is drawn from the unconditional multivariate normal distribution ($S(x)$) based on the covariance matrix, which is in turn based on the Gaussian covariance function Eq. (8).

Finally, the unconditional sample $S$ is conditioned upon the known data points. To do this, the difference between $S(X)$ and the interpolated values of $S(X)$ at the observation locations giving $S^*(X)$ is calculated and gives the kriging variation. A

conditional sample $T$ is then obtained by

$$T(x) = Z^*(x) + (S(x) - S^*(x)). \tag{9}$$

The small scale structure of $T$ represents possible random fluctuations, while the overall large scale trend agrees with the kriging interpolation result (Chiles and Delfiner, 1999; Hadavand and Deutsch, 2020). The advantage of the conditional simulation is the ability to overcome difficulties when dealing with sparse data to estimate non-linear and small scale quantities (Hadavand

and Deutsch, 2020). The simulated field includes a more appropriate spatial correlation between the observation points so that we can obtain an idea of possible local points.

## 2.2 Data

The heat flow observations analyzed in this work are located in the Arctic region north of 65°N. Observations are taken from the data sets of Lucazeau (2019) and Colgan et al. (2022). Lucazeau (2019) has introduced a global compilation of published

heat flow observations. Compared to earlier compilations (e.g. Pollack et al. (1993)), significant changes can be found, espe-





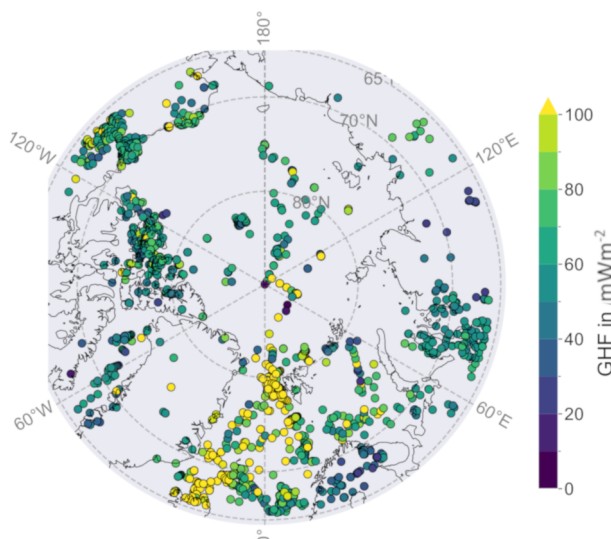

**Figure 2.** Heat flow for the Arctic Region. Data points are from Colgan et al. (2022) and Lucazeau (2019).

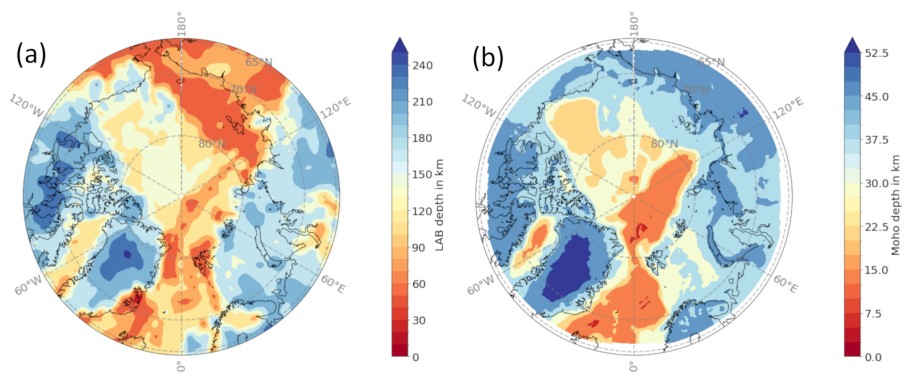

**Figure 3.** Input data: (a) LAB depth from the LithoRef18 (Afonso et al., 2013) and (b) Moho Depth from the ArcCrust model (Lebedeva-Ivanova et al., 2019).

cially for oceanic heat flow, e.g. due to a better quality of sampling in hydrothermal regions. 1488 observations are available for our study area. Colgan et al. (2022) compiled a database for Greenland, which adds 417 more points. The analysis of the heat flow observations is mainly based on these combined 1905 points (Figure 2).

In addition to the heat flow observations we use models for the LAB depth (Afonso et al., 2019) and Moho depth (Lebedeva-Ivanova et al., 2019) for our calculations in order to make statements about the reliability to the Heat Flow points in relation to these Solid Earth models (See Figure 3).



The LAB depth is derived from a joint inversion of gravity anomalies, geoid height, satellite-derived gravity gradients and. constraints from seismic, thermal and petrological data used (Afonso et al., 2019). As Moho depth we use the ArcCRUST

model, which is calculated from 3D forward and inverse gravity modeling with constraints from sediment thickness, rifting age, density and oceanic lithosphere age (Lebedeva-Ivanova et al., 2019). These models and databases are among the most recent available for the Arctic.

For the regional analysis of continental Greenland, 47 heat flow observations are used from the Colgan et al. (2022) database which are located on Greenland or directly on the coast and extend down to 60°N.

## 3 Results

### 3.1 Agreement to Solid Earth Models

At each heat flow point, an ensemble of possible thermal parameter results from the MCMC approach is calculated where we use the mean at each location as the most likely result. The estimated mean parameters (Fig. 4, main diagonal) cover the entire range of allowed values (Table 1).

Both mantle and crustal thermal conductivities show a bimodal distribution with each a peak at the upper boundary and the middle of their prior range. The crustal heat production is distributed nearly uniformly again with a peak at the upper boundary. The peaks at the upper boundary of the parameter range are mostly due to points with a bad fit of the LAB temperature. Other points where the parameters also fall at the edge of the range might also be problematic although a fit to the LAB temperature was possible. Moho and LAB depth are separated into two depths representing the continental and oceanic parts.

When looking at the correlation plots at the lower triangle we see a high correlation between the surface heat flow $q_0$ and the mantle heat flow $q_D$. In general, there is a positive correlation between the fitted parameters and $q_0$ and $q_D$, respectively and a slightly negative correlation between the depths and the surface heat flow and mantle heat flow.

The distribution of thermal parameters (Fig. 5) highlights important spatial trends and underlines where we have no fit to the LAB temperature. For all of the parameters we get values at the upper range for most of the points in the oceanic parts and

some on continental lithosphere, mostly coinciding with a bad fit of the LAB temperature (compare to Fig. 6). Other points with similar values for the thermal parameters might be considered as a problematic fit. It is possible to resolve these points but parameters at the edge of the ranges are less likely.

Despite these points, we see a trend towards the mean values from the possible parameter range for crustal (a) and mantle (b) thermal conductivity, probably indicating that they are not well resolved. This is already seen in the density of the parameter

combination in Fig. 4, e.g. in relation to the Moho depth. The distribution of the crustal heat production tends to follow the GHF with higher values at oceanic lithosphere and lower values and continents. We also find low crustal thermal conductivity at the region of Scandinavia which could also give a hint on problematic input parameters, e.g. too shallow LAB depths. The calculated mantle heat flow follows the LAB depth and is therefore higher in oceanic lithosphere and lower in continental lithosphere. Somewhat paradoxically, the standard deviations from the MCMC runs are very small (Supplementary Material 1)

at the locations where we could not fit the temperature profile, caused by the result clinging to the boundary of the parameter





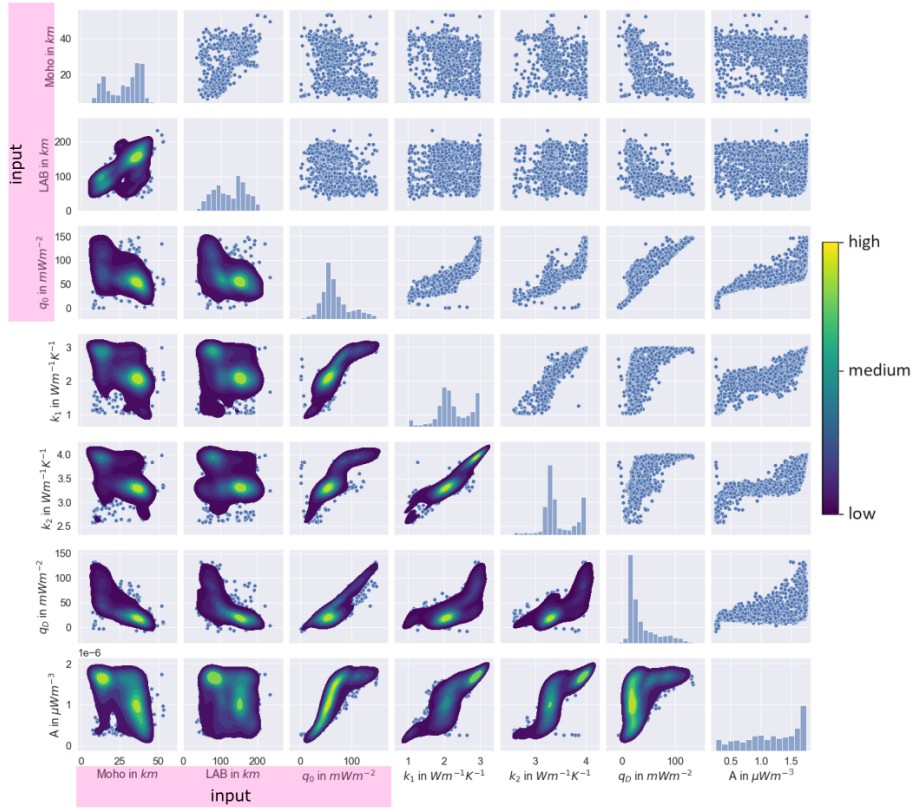

**Figure 4.** Correlations between the input parameters Moho and LAB depth and surface heat flow and the output parameters from the inversion crustal thermal conductivity $k_1$, mantle thermal conductivity $k_2$, mantle heat flux $q_D$ and crustal heat production $A$. The upper triangle shows the inversion results as scatter plots, on the diagonal we find the histograms for each parameter. The lower triangle displays the density of the parameter combinations. Note that points with a GHF of more than $150\,\mathrm{mWm}^{-2}$ are neglected in this figure for the purpose of clarification.

ranges.

Our analysis shows that about 2/3 of the heat flow points can be fit with Solid Earth models if adequate thermal parameters are selected (Figure 6). However, given the range of parameters that we allow it was impossible to achieve the desired $100\,\mathrm{K}$ threshold for LAB temperature at 628 locations. Most of these points are located in the oceanic lithosphere.

Comparing Fig. 5 and 6 (a), we see that most of the high values for the parameters are at locations where the LAB temperature could not be fitted, so that this strictly linear approach might not be appropriate to resolve especially oceanic lithosphere . Allowing small jumps in the temperature at the Moho leads to a not strictly linear representation of the temperature profile and could improve the fit of the LAB temperature for more heat flow points, e.g. where half space cooling is assumed. By choosing $q_D$ as a free parameter and estimate its value with the MCMC algorithm we can imply this within our inversion. To

include $q_D$ as free parameter to the inversion we use the range based on Lösing et al. (2020) with a minimum of $0\,\mathrm{mWm}^{-2}$, maximum of $200\,\mathrm{mWm}^{-2}$ and a proposal std. dev. of $50\,\mathrm{mWm}^{-2}$. With this, we reduce the number from 628 to 18 heat flow



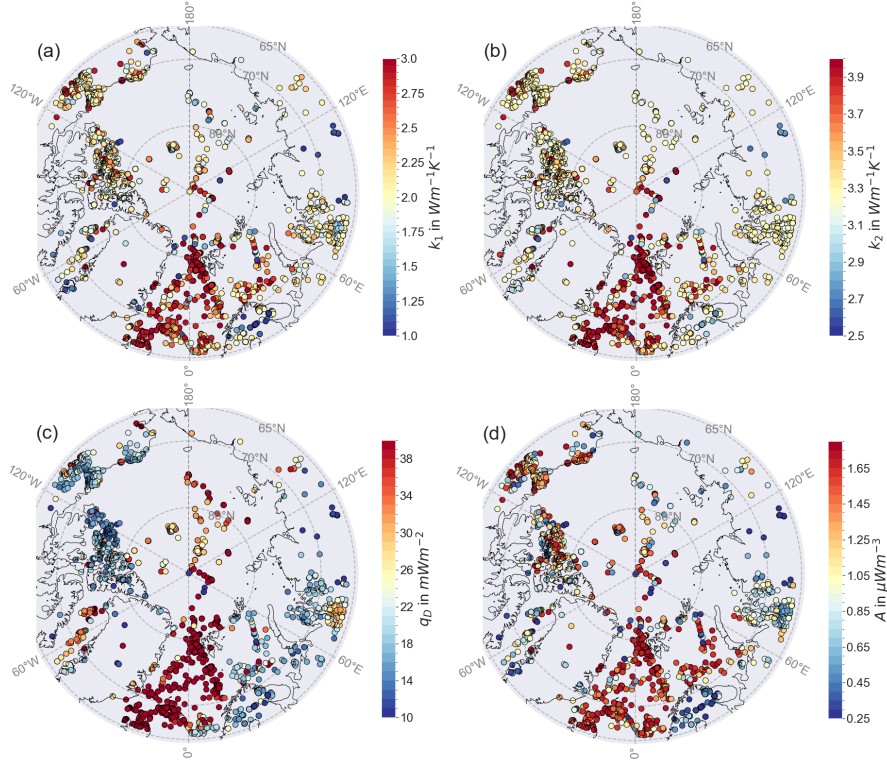

**Figure 5.** Distribution of the thermal parameter from the inversion with $q_D$ calculated with Eq. (3) after 10,000 iterations per point. a) crustal thermal conductivity $k_1$, b) mantle thermal conductivity $k_2$, c) mantle heat flux $q_D$ and d) crustal heat production $A$.

points that do not fit the Solid Earth models and are able to accommodate oceanic points 6 (b). When comparing the approach with $q_D$ as free parameter to other models as the half space cooling model by comparing calculated Curie depths, we mostly get high correlations (Supplementary Material 2). An exception is about 1/6 of the half space cooling comparison where the

Curie depths of the 1D approach tend to be deeper than calculated with half space cooling. So, this approach seems to be robust against changes in parameters. The 18 remaining points are therefore especially interesting since non-linear assumptions still do not lead to a fit. While 17 of the new low reliability points are close to other observations and can therefore be excluded without losing information, the NGRIP point is nearly solitary for central Greenland. Leaving it out leads either to a data gap or high uncertainties in the area of central Greenland when taking the information only from the surrounding points. However,

considering this point within regional studies could be problematic since it appears to not fit the regional geophysical models. For NGRIP, also all parameters lie at the outer edge of the ranges (seen in Fig. 5, discrete values in Table 2) which shows that this heat flow observation of $130\,\mathrm{mWm^{-2}}$ (Colgan et al., 2022) cannot be brought in line with the Solid Earth models using these ranges and preferably should be assumed as local structure or excluded from further studies.



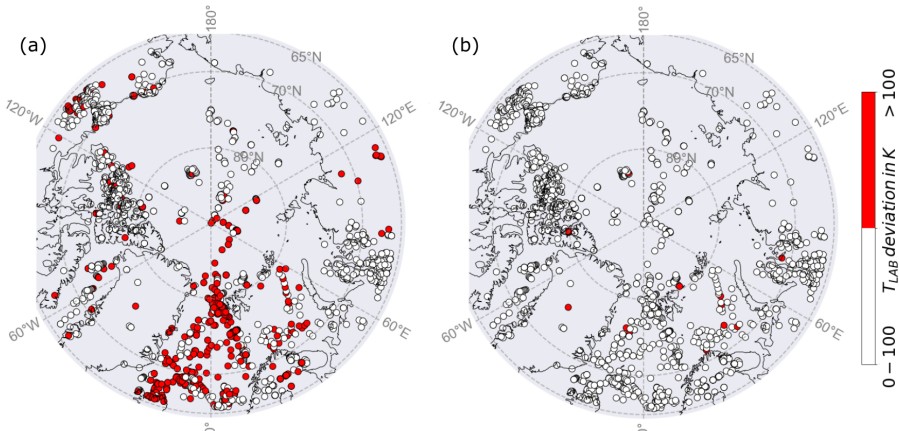

**Figure 6.** Deviation of the calculated compared to the pre-defined LAB temperature (a) with $q_D$ calculated with the crustal heat production $A$ and (b) $q_D$ as a free parameter within the inversion. Most of the points lie within the 100 K uncertainty. For (a) 628 points (red) have a higher deviation, while for (b) the number reduces to 18 points.

**Table 2.** Thermal parameters estimated for the NGRIP point with $q_D$ as a free parameter within the inversion.

|  | $k_1$ in $\mathrm{Wm^{-1}K^{-1}}$ | $k_2$ in$\mathrm{Wm^{-1}K^{-1}}$ | $q_D$ in $\mathrm{mWm^{-2}}$ | $A$ in $\mathrm{\mu Wm^{-3}}$ |
|---|---|---|---|---|
| range | [1.0, 3.0] | [2.5, 4.0] | [0, 200] | [0.25, 1.75] |
| value | 3.0 | 3.3 | 10 | 1.75 |

## 3.2 Kriging interpolation

Simple Kriging and conditional simulation allows to investigate the influence of isolated points in sparse regions. For computational costs, we limit the analysis to Greenland, but the method could also be applied to other regions. We rely on 47 heat flow observations on Greenland or directly at the coast. Within this data set, NGRIP is the only point that does not show an agreement with the regional Solid Earth model.

Semivariogram analysis gives a length scale of 600 km for the whole Arctic. There are not enough data points in Greenland to get reliable results from the semivariogram. However, following Fox Maule et al. (2005), smaller length scales could be more reasonable. Additionally, the length scale for heat flow should be similar in geologically similar regions. Since Greenland once was part of Laurentia on the North American plate (Geoffroy et al., 2001) as well as connected to Norway (Mosar et al., 2002), we assume that a similar spatial variability occurs as in other Precambrian shields, specifically Scandinavia or North America (Näslund et al., 2005). Both regions are well covered with heat flow data so a more reliable semivariogram can be estimated. With the Gaussian variogram model we obtain a length scale of 125 km from the Scandinavian data set (Supplementary Material 3), which then can be applied to Greenland (Fig. 7).





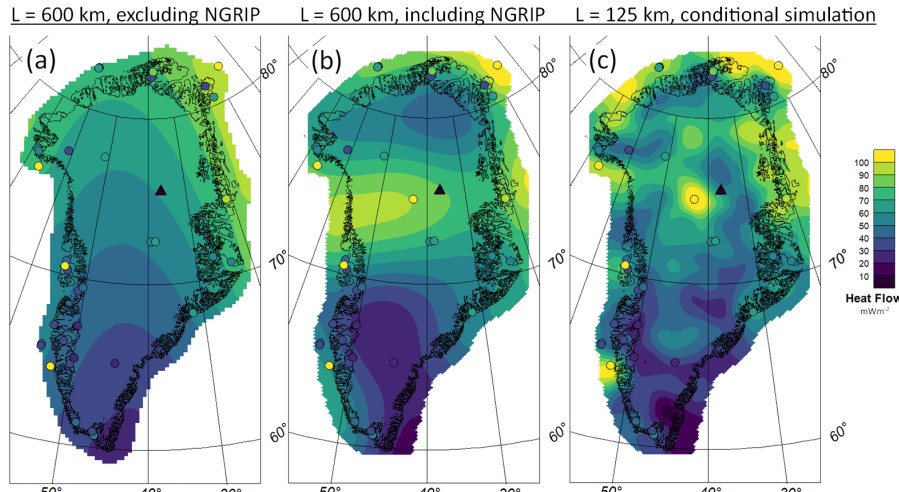

**Figure 7.** Kriging interpolation results for heat flow observations. (a) with a length scale of 600 km excluding the NGRIP point, (b) with a length scale of 600 km including NGRIP and (c) example of a conditional simulation with a length scale of 125 km and NGRIP included.

Carrying out kriging interpolation, we find that, not surprisingly, NGRIP has a crucial impact on the interpolated heat flow field. Leaving it out (Fig. 7 (a)) results in low to medium heat flow values in central Greenland, whereas with NGRIP included (Fig. 7 (b)), lower values are estimated in the north and significantly higher values found in the vicinity of NGRIP, extending c. 300 km west and south. The corresponding uncertainty maps show nearly constant uncertainties of $32\,\mathrm{mWm^{-2}}$ for (a) and $22\,\mathrm{mWm^{-2}}$ for (b) (Supplementary Material 4). With a shorter correlation length of 125 km and applying conditional simulation, a more realistic picture of what heat flow *might* look like, emerges (Fig. 7 (c)). As expected, the reduced correlation length limits the influence of the NGRIP point's high heat flow to a local area. In Southern Greenland - where more points exist - GHF is comparable for all three approaches. Of course, the conditional simulation does not provide any additional constraints on the actual data (Hadavand and Deutsch, 2020).

To further judge the viability of the NGRIP point, we use conditional simulation *without* NGRIP as input. Simulating the unseen local structures in this way, is useful if the stationary heat flow modelling (previous section) implies disagreement between regional geophysical models and measured/inferred heat flow. Using conditional simulation, the statistical distribution of the small scale variations can be probed to assess the possibility of a similarly extreme point occurring. We generate 100 conditional simulations of heat flow without NGRIP to investigate whether heat flow of more than $100\,\mathrm{mWm^{-2}}$ is even possible, with the assumed geostatistical parameters. A circle with 500 km radius around NGRIP will be used as the "vicinity" of NGRIP.

38 % of the simulated heat flow fields exceed $100\,\mathrm{mWm^{-2}}$ in the vicinity of NGRIP. A single realization reached $120\,\mathrm{mWm^{-2}}$, but the reported value of $130\,\mathrm{mWm^{-2}}$ is never attained. Additionally, most simulation have less than 1 % of the NGRIP area vicinity with heat flow values above $100\,\mathrm{mWm^{-2}}$. An area of 1 % is roughly 60 by 60 km, so compared to the length scale this corresponds to a single GHF hot spot in the area around NGRIP. However, in 13 simulations an area of more than 1 % is





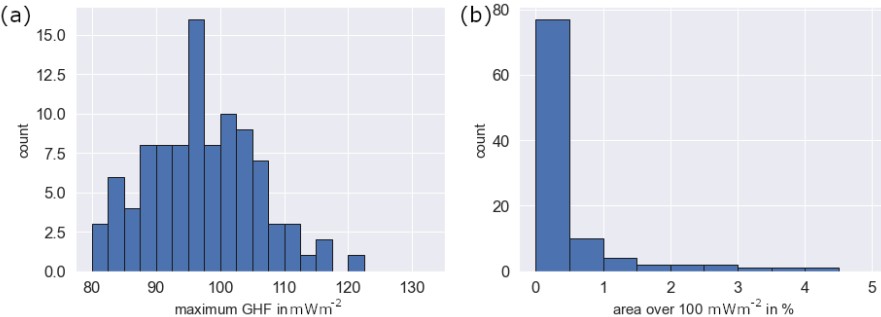

**Figure 8.** Maximum GHF and percentage of GHF over $100\,\mathrm{mWm^{-2}}$ from 100 conditional simulations without NGRIP in the area $500\,\mathrm{km}$ around the NGRIP point.

covered with heat flow values higher than $100\,\mathrm{mWm^{-2}}$, up to almost $5\,\%$ in a single simulation. Within our analysis, $60\,\%$ of the realizations have a maximum heat flow of less than $100\,\mathrm{mWm^{-2}}$ and $87\,\%$ of the realizations have an area of less than $1\,\%$ where $100\,\mathrm{mWm^{-2}}$ are reached. Thus, we can interpret this as a $40\,\%$ chance that there are any "hot spots" above 100 and even

if they did exist, it would be very unlikely (much less than $5\,\%$) that NGRIP randomly "hits" the hot spot. Therefore, the high value of NGRIP cannot be explained with lateral variation at a length scale of $125\,\mathrm{km}$ and would be essentially incompatible with the assumed geostatistical parameters.

EGRIP (East GReenland Ice Project, triangle on map) is a drill site in NNE Greenland without a published heat flow value so far (Rasmussen et al., 2023). This point is close to NGRIP (approximately $190\,\mathrm{km}$) and could provide information on the

spatial influence of NGRIP. Although its heat flow value is not yet published we can still use the location of this point and predict interpolated values for the three different scenarios (Fig. 7). Without NGRIP, EGRIP gets a heat flow of $61\,\mathrm{mWm^{-2}}$. Including NGRIP increases the heat flow at EGRIP to $81\,\mathrm{mWm^{-2}}$ so that we see an influence of the high heat flow of NGRIP. The conditional simulation example gives the EGRIP heat flow at $59\,\mathrm{mWm^{-2}}$ which is significantly lower than the NGRIP value. Performing 50 conditional simulations with NGRIP, we get a variety of possible values for EGRIP (Fig. 9).

In these 50 simulations, the heat flow for EGRIP varies from 40 to $110\,\mathrm{mWm^{-2}}$ with a mean and median of $75\,\mathrm{mWm^{-2}}$. Most of the simulated GHF values for EGRIP lie within the range of 65 to $85\,\mathrm{mWm^{-2}}$.

### 3.3 Basal Melt Estimates

Although our models can be deemed unrealistic, we like to explore shortly the importance for basal melt rates following the approach from Karlsson et al. (2021) (Fig. 10). This shows the effect, local heat flow structures (Fig. 10 (c)) might have on

basal melts compared to two different regional heat flow maps (Fig. 10 (a) and (b)) with an estimated geothermal basal melt for Greenland of $4.9\,\mathrm{Gt}$ per year for local structures and $5.0\,\mathrm{Gt}$ for regional structures.

All of our maps provide similar results for the basal melt (Table 3) with insignificant variations within the single areas. It can be seen that the basal melt for a regional GHF map also shows a regional pattern following the geothermal heat flow while we





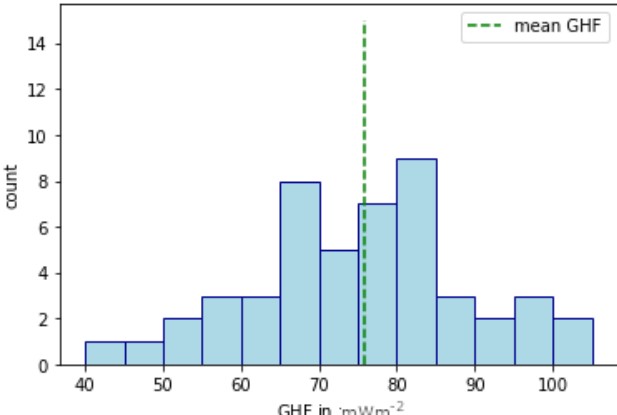

**Figure 9.** Heat flow values for EGRIP extracted from 50 conditional simulations with NGRIP.

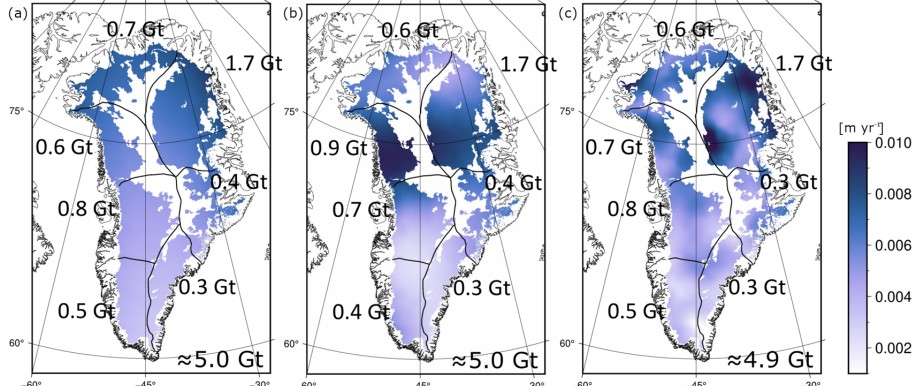

**Figure 10.** Basal melt estimates for Greenland based on the Kriging interpolated heat flow map (a) without NGRIP and 600 km correlation length, (b) with NGRIP and 600 km correlation length and (c) the heat flow map from the conditional simulation with 125 km correlation length and NGRIP. Blanked out areas are considered to be frozen at the ice-bed interface.

see local spots of high basal melt where we have "hot spots" of GHF within the local scale map. Karlsson et al. (2021) use
an average of 3 GHF maps (Fox Maule et al., 2009; Shapiro and Ritzwoller, 2004; Martos et al., 2018) and calculate a total
geothermal basal melt of 5.3+2.8/-2.2 Gt. Basal melt calculated from our HF maps is slightly below but still would be within
the standard deviation.

The largest contribution to basal melt from our GHF maps with 1.7 Gt comes from NE Greenland where the NGRIP point
and therefore the "hot spot" around NGRIP is located. Excluding NGRIP leads to the same basal melt rate for this region.
Karlsson et al. (2021) provide an estimate of 1.3+0.6/-0.5 Gt for this region so that our estimates are slightly higher but within
the standard deviation. The major difference between our estimates and the estimate from Karlsson et al. (2021) can be found
in southern Greenland where Karlsson et al. (2021) estimate significantly more basal melt than our estimate.





**Table 3.** basal melt rates in Gt per year for Greenland for four different HF maps.

|  | Karlsson et al. (2021) | this study, 600 km, no NGRIP | this study, 600 km, NGRIP | this study, 125 km, NGRIP |
|---|---|---|---|---|
| NO | 0.4 ± 0.3 | 0.7 ± 0.3 | 0.6 ± 0.2 | 0.6 |
| NW | 0.6 ± 0.2 | 0.6 ± 0.3 | 0.9 ± 0.2 | 0.7 |
| NE | 1.3 +0.6/-0.5 | 1.7 ± 0.8 | 1.7 ± 0.6 | 1.7 |
| CW | 0.7 +0.5/-0.3 | 0.8 ± 0.5 | 0.7 ± 0.4 | 0.8 |
| CE | 0.5 +0.5/-0.3 | 0.4 ± 0.3 | 0.4 ± 0.2 | 0.3 |
| SW | 1.2 ± 0.4 | 0.5 ± 0.4 | 0.5 ± 0.3 | 0.4 |
| SE | 0.7 +0.5/-0.3 | 0.3 ± 0.3 | 0.3 ± 0.2 | 0.3 |
| total | 5.3 + 2.8/-2.2 | 5.0 ± 2.8 | 5.0 ± 2.1 | 4.9 |

## 4  Discussion

We performed two analyses to appraise the spatial influence of the heat flow observations: First, we used 1-D stationary
heat flow modelling to assess the compatibility between heat flow measurements and regional geophysical models of crustal
thickness and LAB depth. Second, we focussed on Greenland and relied on two related geostatistical techniques to investigate
the impact of the enigmatic NGRIP point on the inferred heat flow.

We find that most of the heat flow observations in the Arctic and Greenland can be made compatible with Solid Earth models,
at least when allowing non-stationary heat flow at the Moho boundary. However, the stationary model fails consistently in the
oceanic domain, particularly in young oceanic lithosphere. This is not surprising, since freshly formed oceanic lithosphere is
cooling rapidly and far from stationary conditions.

We allow wide ranges for the geothermal parameters. Therefore, our quality criteria are fairly lenient and attention should
be focussed on the heat flow points that are incompatible with the geophysical models. Non-agreement could be due to four
reasons: (i) our thermal model might not be adequate for this point, (ii) it could be a measurement error, (iii) the geophysical
models are incorrect or (iv) the measurement is affected by local anomalies. Thus, incompatible heat flow observations should
be used with caution for regional studies, as they could represent local anomalies due to local crustal heat production (Bons
et al., 2021; Hasterok and Chapman, 2011) or hydrological processes, or measurement errors. In Scandinavia incompatibility
is probably due to an incorrect LAB depth, highlighting how our method can also be used to scrutinize the geophysical input
models.

Deciding between the four different reasons is difficult, but the spatial distribution can be helpful. For example, in the case of
the incompatible points in Northern Scandinavia, there is a clear spatial correlation between incompatibility and an unusually
shallow LAB depth. Likewise, any clustering of incompatible points suggests systematic issues rather than local anomalies or
measurement errors. However, ultimately additional (geophysical) data will be needed to clearly determine the reason for the





incompatibility.

Our second analysis is based on geostatistics. The NGRIP point is our particular focus, since it controls the interpolated heat flow over most of central Greenland, as the next heat flow observation is about $300\,\mathrm{km}$ away. A thorough assessment of this point is essential due to its impact on ice sheet modeling (Rogozhina et al., 2016).

We perform jackknifing for NGRIP to test its influence on the length scale the available data for whole of the Arctic provides. Simple Kriging interpolation with poor data coverage lead to high uncertainties (Chiles and Delfiner, 1999), which reach up

to about $32\,\mathrm{mWm^{-2}}$ when NGRIP is excluded from the interpolation data set. Additionally, performing the simple Kriging interpolation with the regional length scales of 600 km confirms that excluding or including a single point can have a large influence on heat flow estimated for central Greenland.

The observation that the GHF at NGRIP is a local effect is supported by the conditional simulation. Furthermore, the NGRIP point has besides the high heat flow value also a high uncertainty, because the value is based on an extrapolated temperature

profile, as it was not possible to drill to the ice-bedrock interface (Dahl-Jensen et al., 2003). An ensemble of conditional simulations shows, that such a high GHF as found at NGRIP is not realistic for central Greenland and only about 10 % of the simulations reach values higher than $110\,\mathrm{mWm^{-2}}$. This leads to the assumption that the estimated GHF value for NGRIP might be too high, assuming the geostatistical parameters are correct. With the help of the statistical analysis, problematic points can be found and excluded from regional studies. We may lose some heat flow observations, but the quality of interpolations and

machine learning can be improved as these points might not be representative.

There are several studies of GHF in Greenland, that assume large areas of elevated GHF. Martos et al. (2018) infer the GHF from the Curie depth calculated from magnetic data while assuming constant thermal conductivity ($2.8\,\mathrm{Wm^{-1}K^{-1}}$) and heat production ($2.5\,\mathrm{\mu Wm^{-3}}$). They predict an area of elevated GHF for NW-SE Greenland. Leaving out points with a without a fit, we get thermal conductivities of $2.25\,\mathrm{Wm^{-1}K^{-1}}$ or lower for Greenland. The crustal heat production varies between 0.5

and $1.25\,\mathrm{\mu Wm^{-3}}$. For both parameters we estimate values that are below the assumed constant values Martos et al. (2018) use. Especially the constant heat production they assume exceeds our range for the crustal heat production by far. Artemieva (2019) uses a thermal isostasy model based on seismic Moho depth data, topography and the assumption that isostatic anomalies can be translated in LAB depth topography. In the region of CE Greenland anomalously high GHF of $110\,\mathrm{mWm^{-2}}$ is calculated that extends with high heat flow cross Greenland. According to our analysis such high heat flow would not influence such

a large area onshore as predicted in the model. The region around NGRIP shows elevated GHF of up to $75\,\mathrm{mWm^{-2}}$ which could be reasonable. Other models such as the machine learning approach in Colgan et al. (2022) suggest that NGRIP point is incompatible with their geophysical data sets. A machine learning model without this point results in no elevated GHF for central Greenland. This is in line with the results from Kriging and the conditional simulations. This confirms that NGRIP should be used with caution for regional studies.

Calculating the basal melt from our GHF maps, we find that the general basal melt is similar to calculations with regional heat flow models. Our local NGRIP structure seems to punctually provide more basal melt. As stated in McCormack et al. (2022) local "hot spots" could have a significant influence and not considering those structures could lead to underestimating the basal melts. When considering a local hot structure we get similar basal melt as choosing the same mean heat flow for Greenland





without local structures. Within the single areas the basal melt varies between the different models so that these local structures from the heat flow give local structures with high basal melts. Such local high basal melt rates could contribute to the sliding of ice shields. Our models get higher basal melt for northern Greenland and lower basal melt for southern Greenland compared to the map calculated from Karlsson et al. (2021). The regional mean of heat flow within these areas has a high influence on the distribution, so that evaluating the reliability of the high heat flow at NGRIP at least is important for basal melt calculations. Local points with basal melts can also contribute to sliding of ice shields. Also, the area with the highest difference between the heat flow maps is found at a region not included in the calculations for basal melts since radar data clearly show that there is no basal melt at the blanked out regions. This could be changed in future so that the contribution of these regions could be large.

In order to verify local GHF structures, local information such as magnetic data or radar data should be included in the calculation in addition to more direct GHF observations such as the not yet published EastGRIP point. Furthermore, heat flow modelling could be improved by including the temperature at the top of the bedrock, derived by ice temperature profiles from Yardim et al. (2021) and Løkkegaard et al. (2022). Using the different HF maps in connection with the ice temperature profiles within the 1D stationary HF equation could provide information on the reliability of the HF maps.

With this new approach we provide information on the reliability and locality of points and show that assuming smaller length scales is appropriate for Greenland. This approach is also applicable to the global heat flow database for evaluation of data points.

## 5 Conclusions

With our statistical analysis we first evaluate whether the heat flow observations in the Arctic region are in agreement with assumptions of the Solid Earth using the 1D stationary heat flow equation and a more flexible solution by setting mantle heat flux as free parameter. A low agreement between Solid Earth and HF point could mean that the heat flow observation shows a local structure e.g. due to an anomaly high crustal heat production but could also show that the GHF observation or other input data contain errors. We find that few HF observations do not fit the regional assumptions of this approach therefore are not trusted for regional studies and should be used with caution for interpolations or machine learning approaches. Heat flow observations are scarcely available throughout the Arctic and are distributed unevenly. Due to this, large length scales are assumed making interpolations on a local scale difficult. Additionally, single points are influencing large areas, leading to high uncertainties. With the help of conditional simulation, it is possible to incorporate local effects that could not previously be represented in modeling. The conditional simulation helps to investigate the general probability of such high heat flow values and gives a spatial variability for the heat flow which shows that we can limit the influence of single observations that would have a large influence when included to regional studies. With this, we can investigate that the high heat flow value for NGRIP is less probable. If we have such high heat flow, assuming local length scales leads to an exclusively local structure for the NGRIP points and therefore overall lower heat flow values for central Greenland. Here, we apply merely a statistical approach. The reliability of the GHF maps could further be investigated by replacing the constant surface temperature with observations



on ice temperature profiles from radar (Yardim et al., 2021). Further, local geophysical data can be used to obtain information on small scales.

*Code and data availability.* Afonso, J., Fullea, J., Griffin, W., Yang, Y., Jones, A., D. Connolly, J., and O'Reilly, S.: 3-D multiobservable
probabilistic inversion for the compositional and thermal structure of the lithosphere and upper mantle. I: A priori petrological information and geophysical observables,435 Journal of Geophysical Research: Solid Earth, 118, 2586–2617, 2013

Colgan, W., Wansing, A., Mankoff, K., Lösing, M., Hopper, J., Louden, K., Ebbing, J., Christiansen, F. G., Ingeman-Nielsen, T., Liljedahl, L. C., et al.: Greenland Geothermal Heat Flow Database and Map (Version 1), Earth System Science Data, 14, 2209–2238, 2022.

Lebedeva-Ivanova, N., Gaina, C., Minakov, A., and Kashubin, S.: ArcCRUST: Arctic Crustal Thickness From 3-D Gravity Inversion, Geo-
chemistry, Geophysics, Geosystems, 20, 3225–3247, https://doi.org/10.1029/2018GC008098, 2019.

Lösing, M., Ebbing, J., and Szwillus, W.: Geothermal Heat Flux in Antarctica: Assessing Models and Observations by Bayesian Inversion, Front. Earth Sci., 8, https://doi.org/10.3389/feart.2020.00105, 2020.

Lucazeau, F.: Analysis and Mapping of an Updated Terrestrial Heat Flow Data Set, Geochemistry, Geophysics, Geosystems, 20, 4001–4024, https://doi.org/10.1029/2019GC008389, 2019.

## Appendix A: Supplementary Material 1

The standard deviation of the thermal parameters estimation with the inversion are displayed in Fig. A1 for $q_D$ calculated and Fig. A2 for $q_D$ treated as a free parameter. For both calculations, most of the points get a standard deviation about 0.4 to $0.5\,\mathrm{Wm^{-1}K^{-1}}$ for $k_1$ and 0.3 to $0.4\,\mathrm{Wm^{-1}K^{-1}}$ for $k_2$. Extremely low standard deviations are mostly found where we can not fit the temperature profiles and the parameters itself stick to the edges of the range. With $q_D$ calculated we find low
standard deviations for the crustal heat production $A$, with $q_d$ as free parameter, $A$ gets higher standard deviations. We get highly variable standard deviations for the estimated $q_D$ which mostly correlate with the GHF. Especially the estimation of $k_1$ and $k_2$ seems to be quite uncertain.

## Appendix B: Supplementary Material 2

To verify that our 1D stationary HF approach is appropriate to represent whole of the area, we calculate a reference depth with
different approaches. These approaches include more appropriate assumptions especially for oceanic lithosphere. The plots in Figure B1 show the correlation between the Curie depths calculated with 4 different approaches and the reference model from the 1D stationary HF approach.

(a) For oceanic lithosphere the assumption of purely vertical heat flow is not appropriate. We calculate the Curie depth $z_{\mathrm{Curie}}$
based on the half space cooling model which is the standard approach for oceanic lithosphere

$$z_{\mathrm{Curie}}(t) = 2\sqrt{\kappa t} \cdot \mathrm{erf}^{-1}\left(\frac{z_{\mathrm{Curie}} - T_0}{z_{\mathrm{LAB}} - T_0}\right) \tag{B1}$$



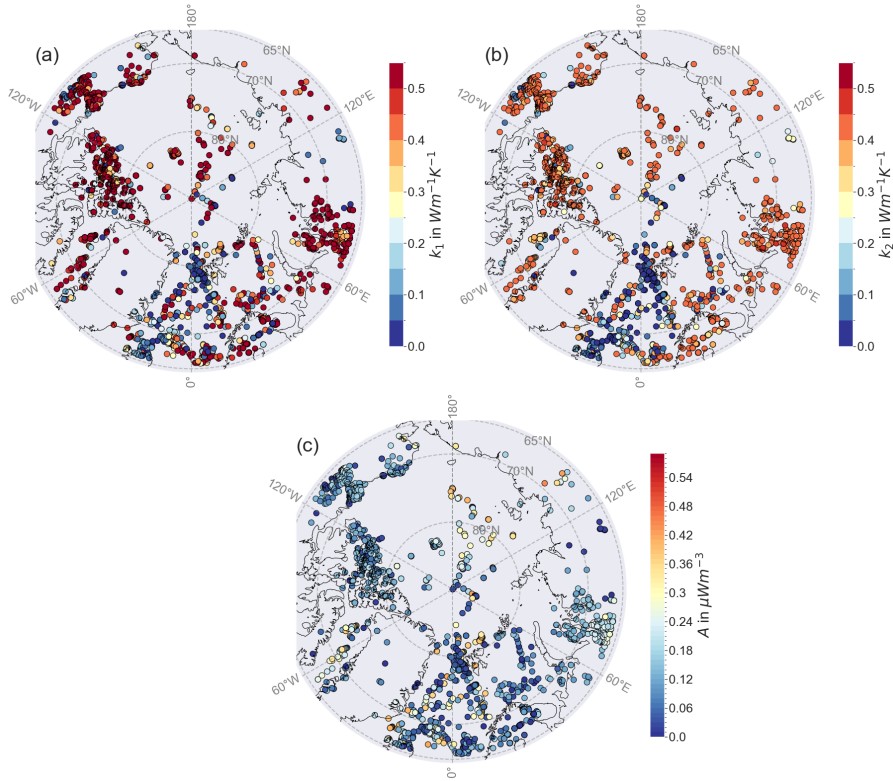

**Figure A1.** Distribution of the standard deviation of the thermal parameters with $q_D$ calculated with Eq. (3). (a) crustal thermal conductivity $k_1$, (b) mantle thermal conductivity $k_2$ (c) crustal heat production $A$.

with the time $t$, Temperatures $T$ for the Curie and LAB depth and the surface and the thermal diffusivity $\kappa = 1.5 \cdot 10^{-6}\,\mathrm{m^2 s^{-1}}$ (Beardsmore and Cull, 2001). Despite the different approach we find a high correlation between the half space cooling approach and the 1D stationary HF approach for shallow Curie depths.

(b) For Greenland, we can compare our reference Curie depth to a Curie depth calculated with given values from the Colgan et al. (2022) database for the crustal thermal conductivity. Due to a shift to lower Curie depths we get a low correlation. Probably the shift comes from lower crustal thermal conductivities given in the database compared to the estimated ones.

(c) Crustal heat production can be assumed negligible for oceanic lithosphere (Beardsmore and Cull, 2001). Therefore, the range for heat production was set to lower values (A* in Table B1). Changing the crustal heat production does not strongly

change the Curie depth.

(d) Last, we additionally can adjust the range for the crustal thermal conductivity. Based on the Colgan et al. (2022) database the crustal thermal conductivity could be lower than initially assumed. The used values $A^*$ and $k_1^*$ are displayed in Table B1. Combining this with the assumption of lower crustal heat production within the oceanic crust, we calculate a Curie depth. We



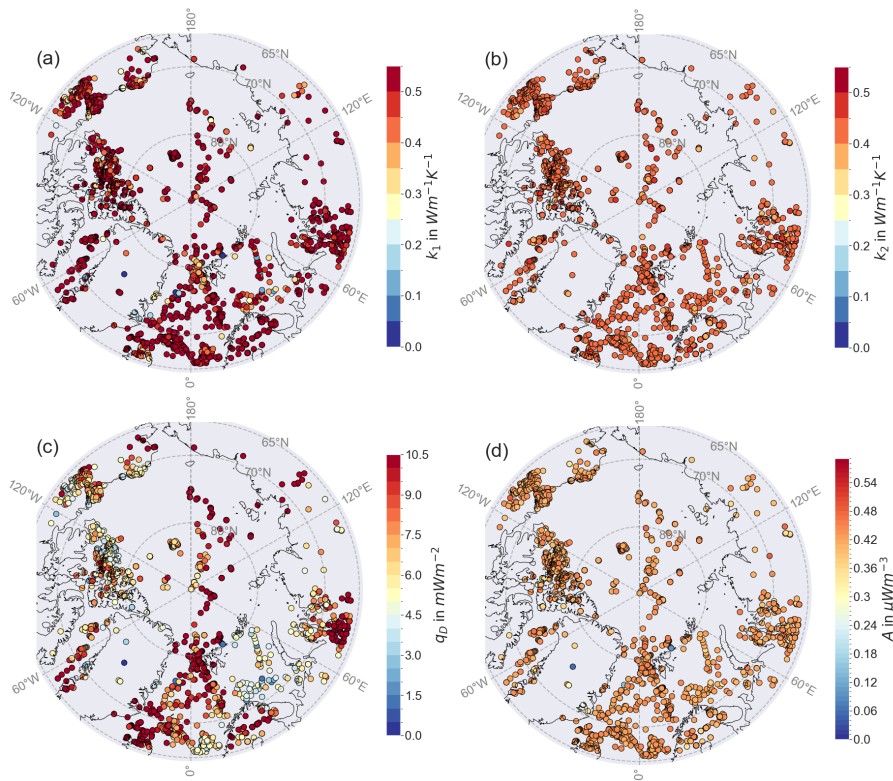

**Figure A2.** Distribution of the standard deviation of the thermal parameters with $q_D$ as a free parameter. (a) crustal thermal conductivity $k_1$, (b) mantle thermal conductivity $k_2$, (c) mantle heat flux $q_D$ and (d) crustal heat production $A$.

**Table B1.** Prior information for the inversion: initial value, range, and step size for each iteration.

| Parameter | initial value | initial range | step size |
|---|---|---|---|
| $k_1$ [Wm$^{-1}$K$^{-1}$] | 2.2 | [1.0, 3.0] | 0.5 |
| $k_1^*$ [Wm$^{-1}$K$^{-1}$] | 2.2 | [0.5, 3.0] | 0.5 |
| $k_2$ [Wm$^{-1}$K$^{-1}$] | 3.0 | [2.5, 4.0] | 0.375 |
| $q_D$ [mWm$^{-2}$] | 22 | [0, 200] | 50 |
| $A$ [μWm$^{-3}$] | 1.5 | [0.25, 1.75] | 0.375 |
| $A^*$ [μWm$^{-3}$] | $1.5 \cdot 10^{-3}$ | [$0.25 \cdot 10^{-3}$, $1.75 \cdot 10^{-3}$] | $0.375 \cdot 10^{-3}$ |

get a good correlation but a higher deviation for the Curie depths compared to the approach (c).



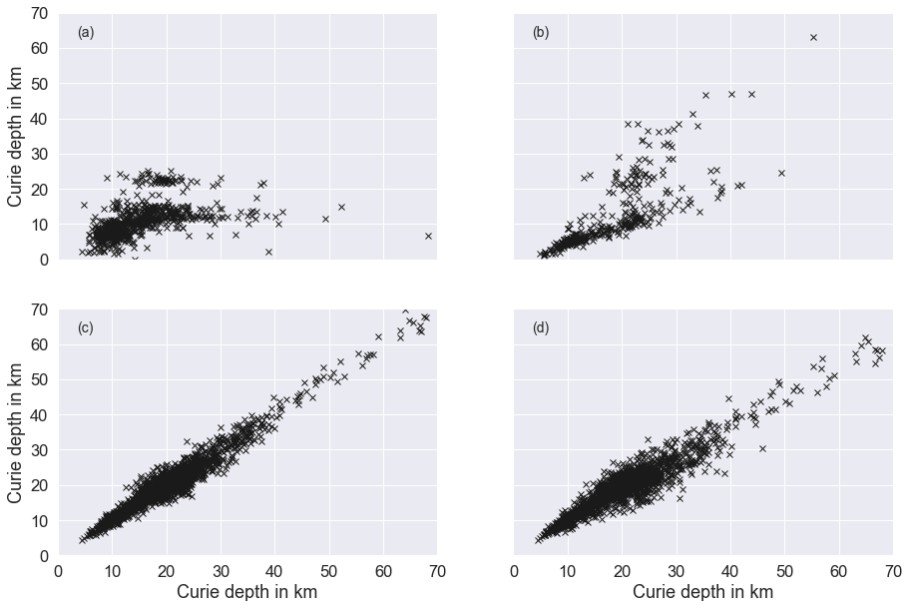

**Figure B1.** Correlation of different Curie depth approaches (y-axis) to the reference Curie depth calculated with the 1D stationary heat flow equation with $q_D$ estimated within the inversion (x-axis). a) half space cooling model (oceanic lithosphere) b) given $k_1$ for Greenland points, c) lower crustal heat production (oceanic lithosphere), d) combination of c) with a lower crustal thermal conductivity $k_1$.

Based on the correlation of the Curie depths we can assume that our approach is appropriate for the analysis. Even for the comparison of oceanic crust we find somewhat high correlations between the different approaches for shallow Curie depths.

### Appendix C: Supplementary Material 3

Due to the geological similarity we take the length scales calculated from the semivariogram for the heat flow in Scandinavia.
Here, we have a high data coverage so that estimates based on the semivariogram are appropriate (Fig. C1). For Scandinavia we so calculate a length scale of 125 km. Comparing the semivariograms (Fig. C2) we see that for Scandinavia we generally get more points with a distribution where the Gaussian variogram model can easily be fitted. For the semivariogram of Greenland we see that we get less points to fit. A fit with the Gaussian variogram model is possible but there are large outliers up to a distance of 500 km so that it is less appropriate to rely on this semivariogram for length scale estimates.

### Appendix D: Supplementary Material 4

Kriging provides the uncertainties for our interpolated heat flow map. For both regional heat flow maps the uncertainty maps are shown in Figure D1. While excluding NGRIP (right) shows uncertainties between 29 and 35 $\mathrm{mWm}^{-2}$, including NGRIP





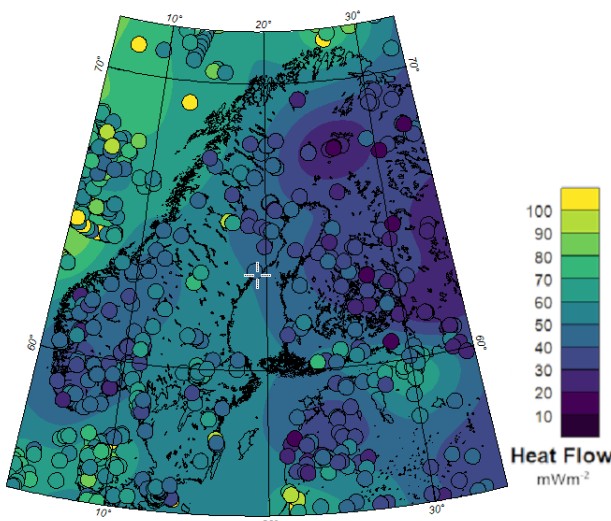

**Figure C1.** Kriging interpolation of the heat flow and observation distribution in Scandinavia.

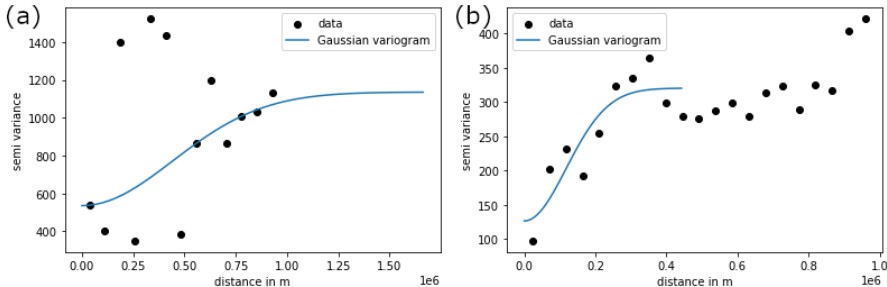

**Figure C2.** Semivariogram for the given heat flow observations for (a) Greenland and (b) Scandinavia.

(left) leads to lower uncertainties of 20 to $25\,\mathrm{mWm^{-2}}$ with higher uncertainties at north-east Greenland. Within both maps we get edge effects.

*Author contributions.* JF drafted the manuscript and performed the calculations (Sect. 3.1 and 3.2). WS contributed to the inversion code. AW performed the calculation of the basal melts (Sect. 3.3). JE contributed to the conception of the study. All authors contributed to manuscript revision, read and approved the submitted version.

*Competing interests.* The authors declare that they have no conflict of interest.





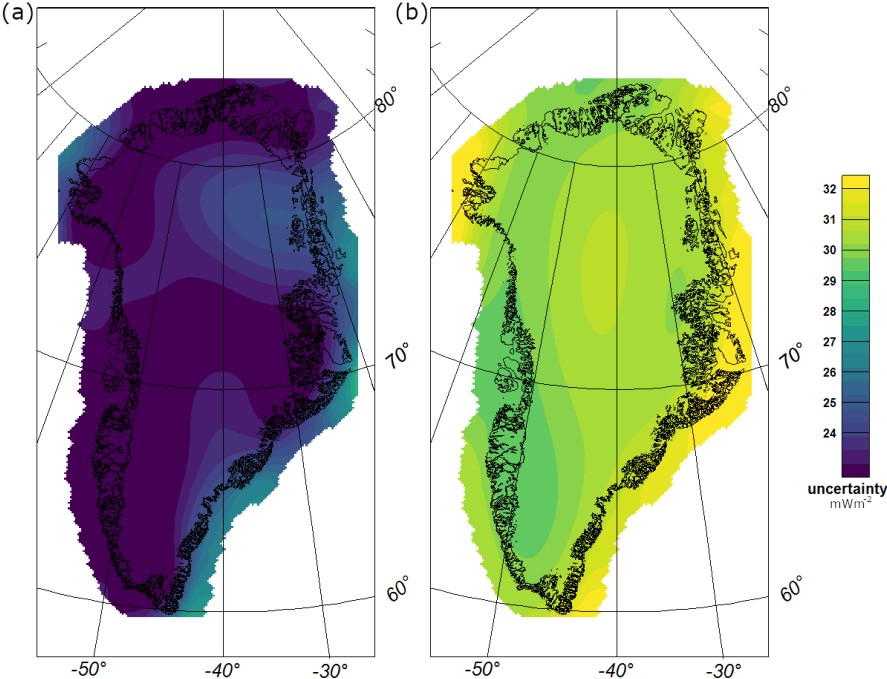

**Figure D1.** Uncertainty maps of the Kriging interpolation (a) with and (b) without NGRIP.

*Acknowledgements.* This work was funded by the Deutsche Forschungsgemeinschaft (DFG, German Research Foundation) - project Green-Crust (459524577) and as a Support to Science Element by the European Space Agency (ESA-STSE 4D Greenland).



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
