# Peer review of "Statistical Appraisal of Geothermal Heat Flow Observations in the Arctic"

_EGUsphere, 2023_

## Author Response (AR1)

*We thank the two reviewers for the thorough revision, which helped us to improve the manuscript. Below, we outline how we have adopted the comments and we hope that the revised version now fulfils the requirements for publication.*

**Reviewer 1**

The study by Freienstein et al. performed two sets of analyses to appraise the quality, consistency and spatial variations of heat flow data in the Arctic region north of 65°N. The data base contains 1905 observations. Each individual measurement is tested for consistency with a lithospheric temperature model based on estimates of Moho and LAB (Lithosphere-Asthenosphere Boundary) depth from regional models. Then geostatistical analysis and conditional simulation are applied to investigate the spatial scale of heat flow in Greenland. Among the purposes is to evaluate the extent to which data are representative and trustworthy and may be used for interpolation and machine learning on a regional scale or in regions with sparse data. Special emphasis is on an investigation of the impact of the enigmatic NGRIP (central Greenland) high heat flow value on the inferred heat flow field. The study processes a large amount of heat flow observations and contains valuable assessment of variations and quality. The study contains valuable results and, in my opinion, deserves to be published, but after some important improvements as suggested below. (Moderate revision).

*We thank the reviewer for their moderate revision on the manuscript and the suggested improvements.*

It is clear that the applied steady state modelling, not surprisingly, does not give good results for oceanic lithosphere. The modelled values for thermal conductivity and heat production in the crust seem unrealistically high, and there is a poor fit to LAB temperature (Fig. 5). For the oceanic lithosphere, the half-space cooling model is a much better model, and it helps to include the Moho/mantle heat flow as a free parameter (Fig. 6). Consequently, I clearly suggest to including that part (Supplementary material 1 and 2) into the main modelling and into the main text with similar weight as for the steady state modelling and with additional explanations. Those parts (Supplementary material) are described only fragmentarily.

*We agree that the thermal parameter values might be unrealistically high for oceanic lithosphere. Our method still is applicable to some extent and to demonstrate this, we included Supplementary Material 1 and 2. Not surprisingly, the stationary steady state approach is inappropriate for (especially young) oceanic lithosphere. To partially overcome this, we propose to add $q_D$ as a free parameter to allow small bends within the temperature profile. Allowing a jump of heat flow at the Moho boundary acts as crude approximation of the non-stationary conditions in oceanic lithosphere. With this adapted approach, more realistic estimates of the thermal parameters are obtained. We can further compare the estimated Curie depths fixed and free $q_D$ with the half space cooling model. In young oceanic lithosphere the differences between the approaches are significant and there is a systematic bias towards deeper Curie depths in our heat flow modelling results. Still, most Curie depths calculated with*

*free qD are close to the half-space cooling Curie depths. In older oceanic lithosphere, our heat flow modelling tends to underestimate Curie depth compared to the half-space cooling expectation. Despite the deviation, we see a similar trend for the different Curie depths in oceanic lithosphere. So, to further evaluate oceanic lithosphere a more advanced model should be considered, which is beyond the scope of our current contribution.*

*In line 210ff, we include a statement on unrealistic high values at oceanic lithosphere. We included the absolute values for the thermal parameters estimated with free qD and the comparison of the Curie depths into the main text (line 237ff + Figure 6 and 7).*

I would like to see a comment on the number of unknowns and constrains on unknowns (thermal conductivities, heat production and mantle heat flow) in the inverse modelling in relation to the possibility of obtaining realistic distributions of parameters when fitting LAB temperatures, and what about trade-off between parameters, which is not mentioned. Are there too many unknowns in relation to the number of known parameters?

*The inverse problem is indeed under-determined. Our main known values are the LAB depth and the heat flow (plus the surface temperature and Moho depth) while the thermal parameters are unknown – There are three when qD is fixed or four unknowns when qD is included as a free parameter. The prior range of thermal parameters is quite wide, to test whether a given GHF point could conceivably be reconciled with regional estimates (see ll. 311-317). Radiogenic heat production is the parameter with the most direct control on surface heat flow, while the thermal conductivities play a secondary role. This is due to the stationary heat flow modelling approach: Temperatures are fixed at the top and the bottom, which limits the impact of heat conductivity. In fact, the posterior distribution of heat conductivity is essentially uniform, except at pathological points. Keeping conductivities free instead of fixed at, say, a mean value, is still preferable, because it provides additional 'wiggle-room' for the inversion and leads to a more conservative (higher) uncertainty estimate. In the future, In-situ measurements of thermal rock parameters could provide tighter constraints on the thermal parameters but this is beyond the scope of this paper.*

There are quite a few examples where sentences lack clarity as well as a number of examples where linguistic syntax is lagging. Should be carefully checked.

*We carefully checked grammar and syntax.*

Reviewer 2

The manuscript centres around the validity of one heat flow observation in Central Greenland called "NGRIP" (I assume this is an acronym although it is never defined). To assess its validity, and other heat flow observations in the Arctic, the authors model steady state geotherms using a 1D analytic solution to the heat equation which is cast within a Bayesian framework to assess uncertainty. It is clear from the onset that this NGRIP point is anomalous, so the stochastic thermal modelling to validate/invalidate this point is superfluous. More interesting is the validation of other heat flow points in the Arctic to examine whether they follow steady-state assumption. However, there are some significant issues I have identified in the

manuscript which prevent me from recommending its publication in its present form, although it may be suitable after moderate revisions.

*We thank the reviewer for their comments. We added the explanation for the acronym NGRIP. We disagree with the reviewers' swift dismissal of NGRIP as an obvious anomaly. Given the remoteness and cost associated with each measurement in Greenland, it is crucial to make the most of every measurement. For this reason, the NGRIP point has almost spawned its own subfield of thermal modelling, as evidence by the large number of publications investigating this point. Previous work (Rezvanbehbahani et al., 2017) is fairly sensitive to NGRIP due to its location in central Greenland.*

My main issue is in relation to the results from the stochastic steady-state modelling. For example, the range of crustal heat production rates in oceanic crust is unreasonably high (Fig 5d). How can oceanic crust be so radiogenic? The thermal conductivity of oceanic crust is also much too high. This is partly addressed in line 210 where the authors include qD as a free parameter, after which the authors claim to have obtained more reasonable results (Fig 6), however these are not shown. These results must exist because the standard deviations on each parameter (k1,k2,qD,A) are shown in the appendix (Fig A2), but the absolute values are missing. Since the initial results produce unrealistically high rates of heat production, I recommend the authors replace those with the results that include qD as a free parameter.

*We agree the thermal parameter values are too high for oceanic lithosphere. Our method still is applicable to some extent and to demonstrate this, we included Supplementary Material 1 and 2 and please see detailed answer to a similar comment by reviewer 1.*

*In line 210ff, we include a statement on unrealistic high values at oceanic lithosphere. We included the figures with free qD as well as Curie depth comparison (In line 237ff + Figure 6 and 7).*

Another issue is the restrictive parameter range used in the stochastic modelling: on line 200, "the standard deviations from the MCMC runs are very small at the locations where we could not fit the temperature profile, caused by the result clinging to the boundary of the parameter ranges" — this indicates to me that the range needs to be extended.

*The prior parameter ranges are already very wide. Thus, in areas where the LAB temperature cannot be fit with our 1-D stationary heat flow model there must be other issues, be it on the data or modelling side. From a purely mathematical point-of-view there might be values which would nevertheless correctly predict the LAB temperature. However, even if we allow such extreme parameters, they could not be representative for a larger region and are hence unsuitable for predicting a regional GHF map. This is the main philosophy behind our approach: Using a very lenient criterion (wide prior ranges) we can be sure that the points where our method fails are problematic in some manner. Conversely, some problematic points will pass through our screening, because we allow wide ranges.*

It is a sensible choice to use the prescribed lithospheric thickness in the model from LithoRef18, and Moho depth from ArcCrust. Although I wonder whether the authors could consider some degree of uncertainty in each of these datasets, and whether this would result in a better fit for some steady-state geotherms. The thickness of the lithosphere in LithoRef18

would have required some thermal modelling, so some discussion is needed on the differences between the thermal calculations in that study compared to those in this manuscript and whether they are compatible.

*The uncertainty of the depth is included indirectly via the standard deviation of the LAB temperature fit within the likelihood function. The chosen temperature deviation (of 100 K) corresponds to a few tens of kilometers of depth uncertainty. The LithoRef18 model does not provide uncertainty estimates (Afonso et. Al 2019). The ArcCrust model gives a standard deviation of 3.8 km for the Moho boundary depth.*

*LithoRef18 does use some simple 1-D temperature modelling. However, the LAB depth is not constrained by heat flow, but temperature is used as a proxy for density: Deeper LAB depth implies a reduction of upper mantle temperatures and thereby a more positive gravity anomaly. This relation forms the basis of the inverse procedure used to determine the LithoRef18 model. Furthermore, LithoRef18 uses the same definition of the thermal LAB (1315 °C) and their linear temperature model in the mantle lithosphere is compatible with our assumption of no heat production in the mantle. LithoRef18 uses constant values for the thermal parameters (conductivity and radiogenic productivity), whereas we investigate lateral variations.*

*We added a statement on the uncertainty of the LAB depth in line 107.*

I'm really unclear on how the priors and data described in the methods section relate to the input/output parameters in Fig 4. I suggest the full likelihood function be written out. Also, is Fig 4 showing the correlations between parameters for a single location or all of the heat flow points in Fig 2? I suspect it is the latter, in which case Fig 4 is probably not the most appropriate way to visualise this information because it is mixing spatial information with statistical information. On reflection I think this figure detracts from the manuscript and should simply be removed. The analytic solution of the steady state heat equation in this paper is well known and widely used. So there is no pressing need for the authors to explore the covariance between each variable. One simply can look at the equation and immediately understand how each parameter is related. A more interesting comparison would be between surface heat flow, Curie depth, age of oceanic crust, and lithospheric thickness.

*The figure in question has been moved to the appendix (A1). It is based on spatial information only, correlating the input data (Moho, LAB and surface heat flow) with the inversion output (heat conductivity, heat productivity). For each point, we use the mean of the posterior distribution.*
*We consider the figure helpful to explore how the input data and thermal parameters are related to each other, illustrating the dependency implicitly contained in the analytical heat flow equation. The figure underlines, that the parameters are quite sensitive to the heat flow and less sensitive to the input depths, which is in our opinion helpful for less experienced readers than the reviewer.*

*The figure has been moved from line 190ff to the appendix line 459ff.*

Was oceanic crust treated any different to continental crust? It would be interesting to see where there is disagreement between the oceanic crustal age and surface heat flow. The

authors mention that they cannot acquire a good fit assuming a model of half-space cooling. Perhaps a plate model of lithospheric cooling would be more apt to avoid overestimating oceanic lithospheric thickness in older seafloor? Indeed, there should be considerably more misfit between heat flow calculations (assuming steady-state geotherms) and heat flow data in newly formed seafloor due to advection.

*In this study, we treated continental and oceanic lithosphere equally. Not surprisingly, it is not possible to fit most of the oceanic lithosphere heat flow with the linear steady state stationary approach, but, within our non-linear approach we were also interested whether the remaining heat flow observations on continental lithosphere could be fitted. Therefore, all the points that could not be fitted with both stationary and non-stationary assumptions are even more interesting.*

**Minor points**

Line 2: "Partly high uncertainty" — what does this mean? Please quantify.

*Has been quantified to up to 30 mW/m²*

NGRIP needs to be defined before the acronym on line 5 in the abstract. Since this point is referred to a lot, it would be helpful to label it on the maps.

*The explanation on the acronym NGRIP has been added, as well as labels on the maps.*

Line 110: is there a criterion used to decide when to stop the Markov chain?

*We decided to use the convergence of the likelihood of our inversion as criterion.*

*In Ll. 110ff we add the clarification of the likelihood function and convergence criterion (l 119).*

Line 169: "As Moho depth" — should be "For Moho depth"

*corrected*

Line 212: Missing Figure before 6b.

*corrected*

Table B1. The step sizes for each parameter are so large I am surprised the MCMC algorithm was able to converge. Again, the convergence criteria should be discussed.

*For clarification renamed in „proposal".*
*We included the explanation in the main text (Methods, line 110f) and changed "step size" to proposal (line 125ff)*

---

## Referee Report (RR1)

Revised version.

Review report

The authors have addressed my comments and suggestion. Where desired, the text is expanded and the results described more clearly. They explain where my suggestions are found to lie outside the scope of this study. This I understand, as some would require significant new modelling.
I suggest publication with technical quality control of figures.

Aarhus 22.02.2024
Niels Balling